# High-risk human papillomavirus genotype distribution, cytological abnormalities, and associated factors among Ethiopian women: a multicenter study

Sisay Tadele,[1,2] Amelework Yilema,[1] Belete Woldesemayat,[1] Kidist Zealiyas,[1] Abinet Admas,[3] Tamrat Abebe,[2] Ayele Gebeyehu,[1] Gemechu Tadesse,[1] Getachew Tollera,[1] Nega Berhe,[4] Nigatu Kebede[4]

**ABSTRACT** Persistent infection with high-risk human papillomavirus (hr-HPV) leads to cervical cancer (CC), the second most common cancer among women in Ethiopia. Understanding the genotype distribution of hr-HPV is essential for effective prevention strategies. However, existing studies in Ethiopia are fragmented and lack comprehensive national evidence. This study assessed the distribution of hr-HPV genotypes, cytological abnormalities, and their associated risk factors in women aged 30–65 in Ethiopia. A facility-based cross-sectional study was conducted from May 2024 to March 2025, including 735 women. Cervical samples were collected for cytological examination and multiplex PCR analysis, and all women were screened with visual inspection with acetic acid (VIA). Data were entered into EpiData version 4.7 and analyzed using SPSS version 27. Overall hr-HPV positivity was 16.2% (95% confidence interval: 13.6–19.2). HPV-16 was the most prevalent genotype (4.8% of all women and 29.4% of hr-HPV cases), followed by HPV-18 and HPV-35. VIA positivity rate was 13.3%. Abnormal cytology was found in 5.7% of women, of whom 40.5% had hr-HPV, while hr-HPV infection was present in 14.7% of women with normal cytology. Among hr-HPV cases, 65.6% of HPV types were not targeted by the current vaccine, and 30.9% of these non-targeted genotypes were detected in women with abnormal cytology. Having multiple sexual partners (aOR: 2.42), early sexual debut (aOR: 2.12), and a history of sexually transmitted infections (aOR: 1.81) were significantly associated with hr-HPV infection. High-risk HPV infection is high in the studied population in Ethiopia, with a significant proportion of genotypes not covered by the current vaccine. Adopting a nonavalent vaccine with sexual health education and regular screening is critical for comprehensive CC prevention.

**IMPORTANCE** The importance of this study lies in generating a comprehensive national evidence on high-risk human papillomavirus (hr-HPV) across multiple sites among Ethiopian women, addressing a significant gap in public health data. It documents the considerable burden of hr-HPV infection, identifies the most prevalent circulating genotypes, and reveals that a large proportion of infections are caused by genotypes not targeted by the currently used quadrivalent vaccine. This finding has profound implications for vaccine policy and cervical cancer (CC) prevention strategies in Ethiopia. By establishing clear associations between hr-HPV infection, cytological abnormalities, and behavioral risk factors, including multiple sexual partners, early sexual debut, and history of sexually transmitted infections. This study highlights the need for integrated prevention approaches. The evidence strongly supports the adoption of a broader-spectrum nonavalent vaccine, complemented with sexual health education and regular cervical screening programs, to effectively reduce the burden of hr-HPV and prevent cervical cancer in Ethiopian women.

**Peer Reviewer** Paul Nagao, The George Washington University School of Medicine and Health Sciences, Washington, DC, USA

Address correspondence to Sisay Tadele, sisay75t@gmail.com.

The authors declare no conflict of interest.

**KEYWORDS** cervical cancer, cytological abnormalities, genotype distribution, high-risk human papillomavirus, HPV vaccination, Ethiopia

Human papillomaviruses (HPVs) are small, double-stranded DNA viruses that are a common cause of sexually transmitted infections (STIs) worldwide (1). More than 220 HPV types have been identified, of which approximately 40 affect the anogenital tract (1, 2). Persistent infection with hr-HPV genotypes is associated with nearly 99% of cervical cancer (CC) (3). Almost 70% of cases are attributed to HPV-16 and HPV-18 (4). Cervical cancer is the fourth most common cancer among women worldwide, with an estimated 662,044 new cases and 348,709 deaths in 2022, 94% of which occurred in low- and middle-income countries (LMICs) (5). Africa is the continent most severely affected, accounting for 18 of the 20 countries with the highest CC rates (6). According to 2022 estimates, nearly 8,168 new cases and 5,975 deaths are reported annually in Ethiopia, with projections indicating a significant increase to 19,234 cases and 14,206 deaths by 2045 (5).

Globally, the prevalence of hr-HPV among women differs owing to regional socioeconomic, behavioral, and demographic factors. A global meta-analysis estimated that 11.7% of women with normal cytology had hr-HPV, with higher rates found in sub-Saharan Africa (SSA) (24.0%), Eastern Europe (21.4%), and Latin America (16.1%) (7). A study conducted in SSA reported a pooled hr-HPV prevalence of 34% (8). In Ethiopia, recent studies have reported varying prevalence rates of hr-HPV among women: 13.1% in a hospital-based study (9), 21.1% in a community-based cohort (10), and as high as 53.0% in a high-risk group (11), which shows diverse study populations and settings.

The hr-HPV genotypes found in women vary by region. Globally, HPV-16, HPV-18, HPV-31, HPV-33, HPV-52, and HPV-58 are the most frequently detected genotypes in women with normal cytology (7, 12). Studies in sub-Saharan Africa have shown that HPV-16, HPV-18, HPV-35, and HPV-52 are common among women who visit health facilities (8). In Ethiopia, HPV-16, HPV-18, HPV-35, HPV-52, and HPV-56 are the predominant genotypes (13). Persistent infection with these hr-HPV genotypes is strongly associated with abnormal cytological findings (3). An African study reported abnormal cervical cytology in 25.8% of women, while recent studies in Ethiopia found prevalence rates of 12% (9) and 13.7% (14). Several demographic and behavioral factors increase the risk of HPV acquisition, persistence, and CC, particularly in resource-constrained settings such as Ethiopia. These include women's age (9, 11), early initiation of sexual activity (12), history of sexually transmitted infections (15), and having multiple sexual partners (9, 13, 16). Persistent infection occurs among 10–20% of infected patients.

Although patterns of global hr-HPV are well-studied, evidence in Ethiopia is limited, and the few studies conducted are scarce, confined to single facilities or retrospective studies, and lack representation across diverse populations, highlighting the need for a comprehensive, multisite study. To bridge this gap and reinforce Ethiopia's commitment to the WHO's 2030 CC elimination goal, this study assessed the prevalence and distribution of hr-HPV genotypes, their association with cytological abnormalities, and the risk factors for hr-HPV infection.

## MATERIALS AND METHODS

### Study design, period, and setting

A facility-based cross-sectional study was conducted from 1 May 2024, to 30 March 2025, at 11 purposefully selected public health facilities in Ethiopia. The study was conducted in five regions (Amhara, Oromia, Gambella, Central Ethiopia, and Sidama) and in the Addis Ababa City administration (Addis Ababa). The study sites were Debre Berhan Comprehensive Specialized Hospital, Worabe Comprehensive Specialized Hospital, Dessie Comprehensive Specialized Hospital, Jimma Teaching and Referral Hospital, Yirgalem General and Teaching Hospital, Adare General Hospital, Shashemene General Hospital, Gambella General Hospital, Federal Police General Hospital, Felege

Meles Health Center, and Adissu Gebeya Health Center. Health facilities were chosen for their active CC screening services and for having gynecologists and pathologists available for clinical diagnosis and cytology examinations, which are crucial for effective screening and early detection. The sample size was proportionally distributed among the hospitals.

## Population

The study population included women aged 30–65 years (17) based on WHO recommendations, who met the minimum inclusion criteria, attended outpatient departments for routine services at health facilities during the study period, and provided consent to participate in the study.

### Inclusion criteria

The study included women aged 30–65 years who visited health institutions during the data collection period, resided in the study area, and were capable of providing informed consent.

### Exclusion criteria

Women who had sex within 24 h before specimen collection, those who experienced heavy vaginal bleeding, additionally, women with documented cervical abnormalities, a history of hysterectomy, or confirmed previous hrHPV infection, and those who were incapable of giving consent were excluded.

## Sample size determination

The sample size was determined using the formula for a single population proportion, with a 4% margin of error, 95% confidence interval (CI), 20.5% prevalence of hr-HPV (18), and a 10% non-response rate, assuming a design effect of 1.5. The final sample size was 735.

## Sampling technique

The study sites in each region were carefully selected based on the availability of CC screening services, the presence of trained and skilled professionals, and high service volume to maximize the sample size within the entire study period. Eligible outpatients seeking services at selected health facilities were enrolled consecutively until the target sample size was reached.

## Operational definition

### Multiple infections

Refer to participants who were infected with more than one hr-HPV genotype simultaneously, including double, triple, quadruple, quintuple, and sextuple infections. It indicates the presence of more than one hrHPV genotype in a single participant.

## Data collection

Trained midwives conducted face-to-face interviews using a pretested structured questionnaire to collect sociodemographic, behavioral, and clinical data. Pathologists performed cytological examinations, while trained gynecologists and midwives visually inspected the cervix and collected cervical swabs.

## Cervical swab collection and storage

### Cervical swab specimen collection

Cervical swab samples were collected using the PreservCyt solution from Halogics, Inc., following the manufacturer's guidelines for the collection and handling of cervical

specimens in PreservCyt solution. The samples were transported to the Ethiopian Public Health Institute (EPHI) laboratory at a temperature of 2–30°C and stored at −80°C for further processing.

### Visual inspection with acetic acid (VIA)

Women were screened for CC using visual inspection with acetic acid (VIA), and the uterine cervix was examined after 3–5% acetic acid was applied to detect acetowhite changes indicative of precancerous or cancerous lesions. Women with invisible transformation zones or active vaginal bleeding were excluded from the study (19).

### Cytological examinations and interpretation results

Cervical samples for cytological examination were collected using a Cervex-brush (Rovers Medical Devices, Netherlands). Smears were prepared on slides, fixed with ethanol, and stained following standard protocols (20). Each slide was independently examined by two experienced pathologists and analyzed according to the 2014 Bethesda System for reporting pap results. When pathologists disagreed on a diagnosis, they reviewed the slides together, and a consensus was reached. Cytological results were classified as: negative for intraepithelial lesions or malignancies (NILMs), atypical squamous cells of undetermined significance (ASC-US), atypical squamous cells, cannot exclude high-grade squamous intraepithelial lesion (ASC-H), low-grade squamous intraepithelial lesions (LSIL), and high-grade squamous intraepithelial lesions (HSIL) (21).

## HPV DNA extraction, detection, and genotyping

DNA extraction, detection, and genotyping were performed at the EPHI National HIV Reference Laboratory (NHIVRL) using the Abbott RealTime hr-HPV assay (Wiesbaden, Germany). This assay classifies HPV into three genotypes: HPV 16, HPV 18, and non-HPV-16/18 genotypes. Additionally, the Anyplex II HPV HR Detection Kit (Seegene, Seoul, South Korea) was used to further genotype the non-HPV-16/18 samples into 14 hr-HPV genotypes.

The Abbott RealTime hr-HPV assay utilizes the Abbott *m*2000sp instrument for sample processing; extracts, concentrates, and purifies the target DNA molecules using magnetic particles (Abbott *m*Sample Preparation System$_{DNA}$ for RealTime hr-HPV; and the Abbott *m*2000rt instrument for amplification and detection after mixing the extracted DNA with the PCR master mix according to the manufacturer's recommendations. It employs a primer mix targeting a conserved L1 region. An internal control (IC)-specific probe was used to identify the endogenous human β-globin sequence, which was the target of the primer set used to create IC amplicons. This assay utilizes distinct fluorophores to label probes for HPV-16, HPV-18, and non-HPV-16/18 genotypes, and IC, thereby allowing signal differentiation in a single reaction (22).

The Anyplex II HPV HR Detection and Genotyping Kit was used according to the manufacturer's instructions on the CFX96 Real-time PCR System (Bio-Rad, Hercules, CA, USA) to further classify the other hr-HPV genotypes identified using the Abbott platform into 14 hr-HPV genotypes. This experimental setup was used for the detection of hr-HPV using 5 µL of eluted DNA in a total volume of 20 µL. It employs double-priming oligonucleotide (DPO) and Tagging Oligonucleotide Cleavage and Extension (TOCETM) technology, facilitating the accurate detection and quantification of 14 hr-HPV subtypes (16, 18, 31, 33, 35, 39, 45, 51, 52, 56, 58, 59, 66, and 68), incorporating human beta-globin as an IC. Positive and negative controls were included in each reaction (23).

## Quality control

The questionnaire was initially developed based on previously published research articles in peer-reviewed journals, written in English, translated into local languages (Amharic, Sidamigna, and Afan Oromo), and back-translated. The questionnaire was

piloted and revised accordingly. The data and specimen collectors were trained on the data collection procedures, data quality, and study objectives. To determine the cytological characteristics of the slides, two pathologists independently and blindly evaluated the stained slides for their consistency and quality. They both discussed and resolved discrepancies. Laboratory tests were performed according to standard operating procedures. The kit's positive and negative controls were run alongside the tests to validate the entire PCR workflow. The testing laboratory at the NHIVRL of the EPHI received an accreditation certificate from the Ethiopian Accreditation Service (EAS) in 2017, demonstrating compliance with the international ISO 15189:2022 standards.

## Data management and statistical analysis

The data were entered into EpiData version 4.7 and exported to SPSS version 27 for analysis of the data. Descriptive analyses, including frequencies, proportions, and summary measures, were used to characterize the study populations. Variables with a *P* value < 0.25 in the binary logistic regression model were included in the multivariable logistic regression model to identify factors associated with HPV infection. A *P* value < 0.05 in the multivariable model was considered statistically significant.

## RESULTS

### Characteristics of the study participants

A total of 735 women were enrolled in the study, with a mean age of 39.3 years (SD ± 7.13). Among the participants, 46.1% (*n* = 339) were housewives, and 54.8% (*n* = 403) were >15 years old at the time of first sexual intercourse. Most participants lived in urban areas (69.7%, *n* = 512) and were married (80.4%, *n* = 591), with education levels ranging from no formal education (28.0%, *n* = 206) to tertiary education (18.6%, *n* = 137). The majority of participants reported having one lifetime sexual partner (51%, *n* = 375), 52.1%(*n* = 383) reported long-term contraceptive use, and 19.2%(*n* = 141) had a history of STI (Table 1).

### Prevalence and type-specific distribution of hr-HPV infection

Fourteen hr-HPV genotypes were identified, with an overall positivity rate of 16.2% (*n* = 119, 95% CI: 13.6–19.1). Among the hr-HPV positive women, 33.6% (95% CI: 0.25–0.42) had multiple infections, whereas 66.4% (95% CI: 0.57–0.74) had a single HPV genotype (Fig. 1). As shown in Table 2, of the 40 multiple hr-HPV infections, 17.6% had double infections, 4.2% had triple infections, 6.7% had quadruple infections, 4.2% had quintuple infections, and 0.8% had sextuple infections.

In this study, the most common hr-HPV genotype was HPV-16, which was detected in 4.8% of all women and 29.4% of those with hr-HPV (95% CI: 21.0–38.0). HPV-18 and HPV-35 were each found in five of the cases. In addition, HPV-39, HPV-51, and HPV-59 each accounted for 4.2% of the cases. Except for HPV-45, the other hr-HPV genotypes were detected in single infections, ranging from 29% to 0.8%. The prevalence of different hr-HPV genotypes in multiple infections was also analyzed. HPV-51 was detected in six cases, HPV-35 in five cases, and HPV-16, HPV-35, HPV-52, and HPV-56 were each detected in four cases of double infections. In addition, HPV-35 was detected in four of the five triple infection cases. Overall, HPV genotype 51 was the most common hr-HPV genotype found in co-infections with other HPV genotypes.

Similarly, screening with VIA was performed for all study participants. The VIA positivity rate was 13.3% (*N* = 98). Of these, 11.2% presented with cervical epithelial cell abnormalities on Pap smears, whereas 22.4% were hr-HPV positive on PCR.

### Age-specific prevalence of HPV genotypes

The largest proportion of women was in the 30- to 39-year age group (54.8%). Overall, HPV-16 was the most prevalent HPV genotype in almost all age groups (Table

**TABLE 1** Sociodemographic and health-related characteristics of the study participants in Ethiopia, 2025. (*n* = 735)

| Characteristics | *n* (%) |
|---|---|
| Age category (in years) | |
| 30–39 | 403 (54.8) |
| 40–49 | 262 (35.6) |
| ≥50 | 70 (9.5) |
| Residence | |
| Urban | 512 (69.7) |
| Rural | 223 (30.3) |
| Educational status | |
| No formal education | 206 (28.0) |
| Primary (1–8 grade) | 205 (27.9) |
| Secondary (9–12 grade) | 187 (25.4) |
| Tertiary (college and above) | 137 (18.6) |
| Marital status | |
| Never married | 44 (5.7) |
| Married | 591 (80.4) |
| Divorced | 64 (9.0) |
| Others[a] | 36 (4.9) |
| Occupational status | |
| Housewives | 339 (46.1) |
| Government employed | 105 (14.3) |
| Self-employed | 214 (29.1) |
| Student | 43 (5.9) |
| Others[b] | 34 (4.6) |
| Age at first sexual intercourse | |
| ≤15 years old | 332 (45.2) |
| >15 years old | 403 (54.8) |
| Parity | |
| Nulliparous | 62 (8.4) |
| 1–2 births | 434 (59) |
| ≥3 births | 239 (32.5) |
| Long-term contraceptive use | |
| No | 352 (47.9) |
| Yes | 383 (52.1) |
| Lifetime sexual partners | |
| Single | 375 (51) |
| Multiple | 360 (49) |
| History of STI | |
| No | 594 (80.8) |
| Yes | 141 (19.2) |
| Ever screened/tested for HPV | |
| No | 735 (100) |
| Yes | 0 (0) |
| HPV vaccination status | |
| No | 735 (100) |
| Yes | 0 (0) |

[a]Others: Widowed, living together/separated.
[b]Others: Farmers.

2). While single infections were more common, multiple hr-HPV infections were also observed, occurring more frequently in the 40–49 age group (8%, 95% CI: 0.05–0.12). One woman (1.4%) aged 55 years had sextuple hr-HPV infection. Most cervical epithelial

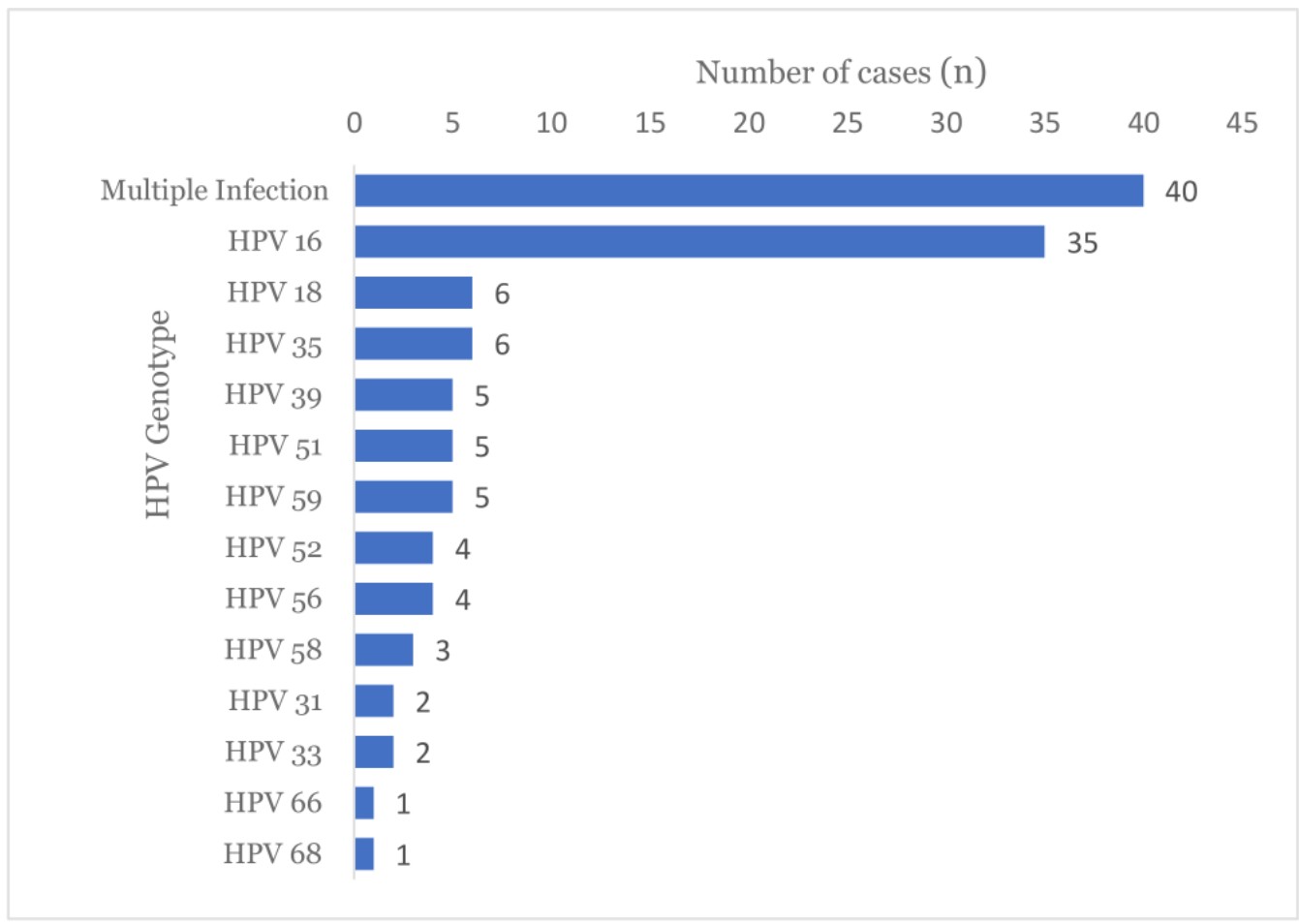

**FIG 1** Distribution of hr-HPV genotypes among hr-HPV positive women in Ethiopia, 2025.

cell abnormalities were found in women aged 40–49 years (47.6%, 20 out of 42 cervical abnormalities) (Table 2).

## Cytological profile of the study participants

Forty-two out of 735 (5.7%, 95% CI: 4.19–7.7) women had cervical epithelial cell abnormalities. The most common findings were atypical squamous cells of undetermined significance (ASC-US), observed in 38.1% (16/42) of the cases, low-grade squamous intraepithelial lesions (LSILs) observed in 33.3% (14/42) of the cases, high-grade squamous intraepithelial lesions (HSILs) observed in 19.1% (8/42) of the cases, and atypical squamous cells cannot exclude HSIL (ASC-H) observed in 9.5% (4/42) of the cases (Fig. 2). Furthermore, 40.5% (17/42) of women with cervical abnormalities were positive for hr-HPV infection, with LSIL accounting for the majority of cases (23.8% [10/17]), followed by HSIL (11.9%, 5/17) (Table 3). Similarly, the hr-HPV positivity rate in women with normal cytology was 14.7% (102/693). Within this group, HPV-16 accounted for 4.6% (32 cases), HPV-18, HPV-39, and HPV-51 each accounted for 0.7% (5 cases each), and dual HPV infections accounted for 2.9% (20 cases).

## Non-vaccine-targeted HPV genotypes and types of infection

The Ethiopian national HPV immunization program recommends and implements a quadrivalent vaccine targeting HPV-6, HPV-11, HPV-16, and HPV-18. However, we identified several genotypes not targeted by the current prophylactic quadrivalent vaccine. Among 119 hr-HPV positive women, 65.6% (95% CI: 0.56–0.73) were infected

TABLE 2 High-risk HPV prevalence and genotype distribution of study participants in Ethiopia, 2025[a]

| Age groups (years) | 30–39 | 40–49 | >55 | Total |
|---|---|---|---|---|
| | n (%) | n (%) | n (%) | n (%) |
| Samples analyzed | 403 (54.8) | 262 (35.7) | 70 (9.5) | 735 (100) |
| HPV positivity | 45 (11.2) | 57 (21.8) | 17 (24.3) | 119 (16.2) |
| Single infections | 31 (7.7) | 36 (21.8) | 12 (17.1) | 79 (10.7) |
| Multiple infections | 14 (3.5) | 21 (8) | 5 (7.1) | 40 (5.4) |
| Double | 9 (2.2) | 9 (3.4) | 3 (4.3) | 21 (2.9) |
| Triple | 3 (0.7) | 2 (0.7) | 0 (0) | 5 (0.7) |
| Quadruple | 1 (0.2) | 6 (2.3) | 1 (1.4) | 8 (1.1) |
| Quintuple | 1 (0.2) | 4 (1.5) | 0 (0) | 5 (0.7) |
| Sextuple | 0 (0) | 0 (0) | 1 (1.4) | 1 (0.14) |
| HPV genotypes | | | | |
| HPV-16 | 16 (3.9) | 12 (4.6) | 7 (10) | 35 (4.8) |
| HPV-18 | 1 (0.2) | 5 (1.9) | 0 (0) | 6 (0.8) |
| HPV-31 | 2 (0.4) | 0 (0) | 0 (0) | 2 (0.27) |
| HPV-33 | 0 (0) | 1 (0.4) | 1 (1.4) | 2 (0.27) |
| HPV-35 | 0 (0) | 3 (1.1) | 3 (4.3) | 6 (0.8) |
| HPV-39 | 3 (0.7) | 2 (0.7) | 0 (0) | 5 (0.7) |
| HPV-45 | 0 (0) | 0 (0) | 0 (0) | 0 (0) |
| HPV-51 | 3 (0.7) | 2 (0.7) | 0 (0) | 5 (0.7) |
| HPV-52 | 1 (0.2) | 3 (1.1) | 0 (0) | 4 (0.5) |
| HPV-56 | 1 (0.2) | 3 (1.1) | 0 (0) | 4 (0.5) |
| HPV-58 | 1 (0.2) | 2 (0.7) | 0 (0) | 3 (0.4) |
| HPV-59 | 2 (0.4) | 3 (1.1) | 0 (0) | 5 (0.7) |
| HPV-66 | 0 (0) | 0 (0) | 1 (1.4) | 1 (0.14) |
| HPV-68 | 1 (0.2) | 0 (0) | 0 (0) | 1 (0.14) |
| Cytology diagnosis | | | | |
| NILM | 390 (96.7) | 242 (92.3) | 61 (87.1) | 693 (94.3) |
| ASC-US | 6 (1.5) | 7 (2.7) | 3 (4.3) | 16 (2.2) |
| LSIL | 5 (1.3) | 8 (3.1) | 1 (1.4) | 14 (1.9) |
| ASC-H | 2 (0.5) | 1 (0.4) | 1 (1.4) | 4 (0.5) |
| HSIL | 0 (0) | 4 (1.5) | 4 (5.7) | 8 (1.1) |

[a]ASC-H, atypical squamous cells cannot exclude HSIL; ASC-US, atypical squamous cells of undetermined significance; HSIL, high-grade squamous intraepithelial lesion; LSIL, low-grade squamous intraepithelial lesion.

with non-vaccine-targeted genotypes, including multiple infections. Thirty-two percent of women (95% CI: 0.23–0.41) had a single non-vaccine-targeted hr-HPV genotype, while 33.6% (95% CI: 0.25–0.42) had multiple infections. Furthermore, HPV-35 was the most common genotype (18.8%), followed by HPV-39, HPV-51, and HPV-59, which were each found in 13.1% of women. Additionally, 30.9% of women with cervical epithelial cell abnormalities were infected with these genotypes. The most frequently detected genotypes were HPV-35 and HPV-58.

## Factors associated with hr-HPV infection among women

In this study, variables with a $P$ value < 0.25 in the bivariate analysis were included in the multivariable logistic regression model. Women aged 40–49 years had significantly greater odds of infection than those aged 30–39 years (aOR = 2.13, 95% CI: 1.34–3.40, $P$ = 0.005). The more lifetime sexual partners a person had, the greater the odds of hr-HPV infection (aOR = 2.42, 95% CI: 1.56–3.77, $P$ < 0.001). Women with a history of sexually transmitted infections had greater odds of infection (aOR = 1.81, 95% CI: 1.11–2.95, $P$ = 0.017). Early sexual debut below the age of 15 years was also associated with higher odds of hr-HPV infection (aOR = 2.12, 95% CI: 1.37–3.28, $P$ =0.001). (Table 4).

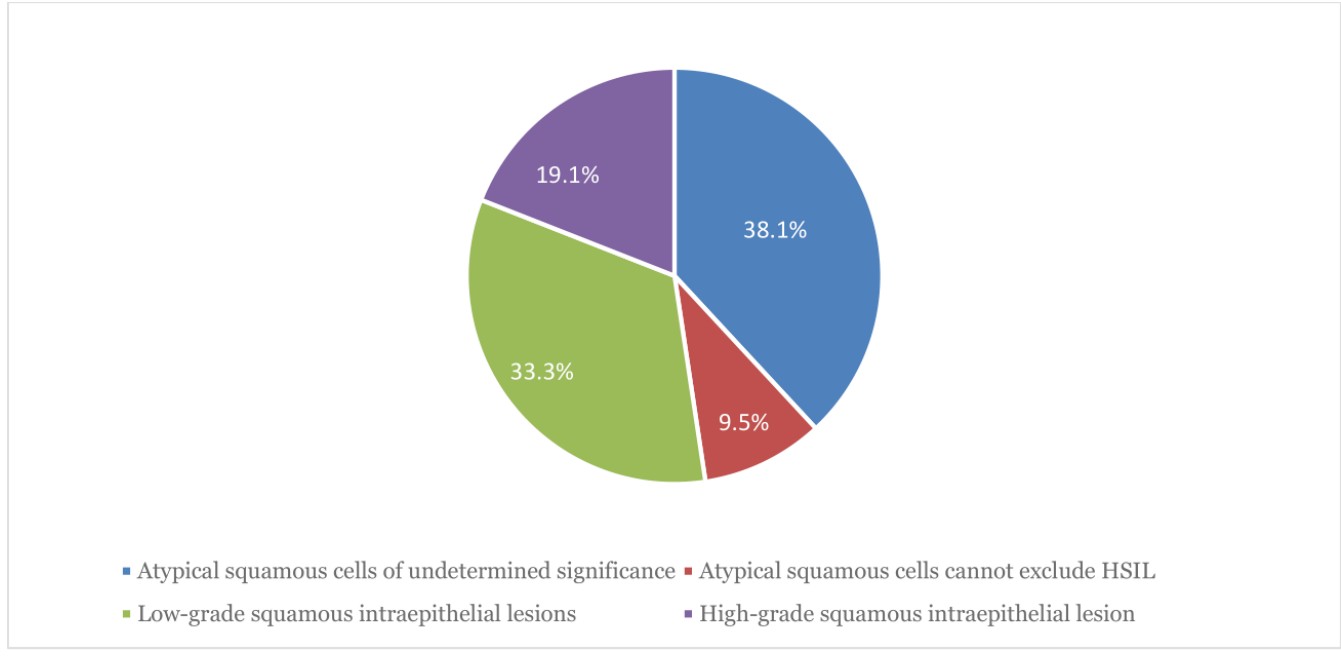

**FIG 2** Cervical cytology profiles of study participants in Ethiopia, 2025.

## DISCUSSION

In this study, the prevalence and genotype distribution of hr-HPV, its association with cervical abnormalities, and related risk factors were explored in 735 women across Ethiopia. Of these, 16.2% were hr-HPV positive. The most common hr-HPV genotypes were HPV-16, HPV-18, HPV-35, HPV-39, HPV-51, and HPV-59. Cervical abnormalities were identified in 5.7% of women, with ASC-US and LSIL being the most common. Multiple lifetime sexual partners, a history of STIs, and early initiation of sexual activity were identified as risk factors associated with higher hr-HPV infection rates. To our knowledge,

**TABLE 3** Distribution of hr-HPV genotypes among women with abnormal cervical cytology findings in Ethiopia, 2025[a]

| Characteristics | Cytology results | HPV detected (n) (%) | Specific HPV genotypes |
|---|---|---|---|
| PAP test | Normal | 102 out of 693 (14.7) | (Listed below with specific |
| | Abnormal | 17 out of 42 (40.5) | categories) |
| Single HPV infection | HSIL | 3 out of 119 (2.5) | HPV-16 |
| | | | HPV-35 |
| | | | HPV-58 |
| | LSIL | 5 out of 119 (4.2) | HPV-16 (in 2 cases) |
| | | | HPV-18 |
| | | | HPV-58 |
| | | | HPV-59 |
| | ASC-US | 2 out of 119 (1.6) | HPV-33 |
| | | | HPV-35 |
| Multiple HPV infection | HSIL | 2 out of 119 (1.6) | HPV-16, -18, -31, -45, -52, -68 |
| | | | HPV-51, -52, -58, -66, -68 |
| | LSIL | 5 out of 119 (4.2) | HPV-56, -66 |
| | | | HPV-16, -18, -39 |
| | | | HPV-16, -18, -51, -52 |
| | | | HPV-31, -35, -39, -58, -59 |
| | | | HPV-16, -35, -39, -45, -52 |

[a]ASC-H, atypical squamous cells, which cannot exclude HSIL; ASC-US, atypical squamous cells of undetermined significance; HSIL, high-grade squamous intraepithelial lesion; LSIL, low-grade squamous intraepithelial lesion.

TABLE 4  Multivariate analysis of factors associated with hr-HPV infection in Ethiopia 2025[a]

| | | HPV DNA result | | COR (95% CI) | AOR (95% CI) | P value |
|---|---|---|---|---|---|---|
| | | Not detected (%) | Detected (%) | | | |
| Age group (years) | 39 | 358 (48.7) | 45 (6.1) | 1.00 | 1.00 | 0.005[b] |
| | 40–49 | 205 (27.9) | 57 (7.8) | 2.2 (1.44–3.39) | 2.13 (1.34–3.40) | |
| | ≥50 | 53 (7.2) | 17 (2.3) | 2.55 (1.36–4.78) | 2.01 (0.98–4.09) | |
| Residence | Urban | 195 (26.5) | 28 (3.8) | 1.00 | 1.00 | 0.056 |
| | Rural | 421 (57.3) | 91 (12.4) | 1.51 (0.95–2.38) | 1.64 (0.99–2.71) | |
| Marital status | Single | 34 (4.6) | 8 (1.1) | 1.00 | 1.00 | 0.55 |
| | Married | 504 (68.6) | 86 (11.7) | 0.73 (0.33–1.62) | 0.82 (0.33–2.04) | |
| | Divorced | 52 (7.1) | 13 (1.8) | 1.06 (0.40–2.83) | 0.89 (0.31–2.59) | |
| | Others | 26 (3.5) | 12 (1.6) | 1.96 (0.70–5.50) | 1.49 (0.46–4.80) | |
| Occupation | Housewives | 292 (39.7) | 47 (6.4) | 1.00 | 1.00 | 0.16 |
| | Govt employed | 92 (12.5) | 13 (1.8) | 0.88 (0.45–1.69) | 0.78 (0.37–1.161) | |
| | Self-employed | 173 (23.5) | 41 (5.6%) | 1.47 (0.93–2.33) | 1.50 (0.90–2.48) | |
| | Student | 33 (4.5) | 10 (1.4) | 1.88 (0.87–4.07) | 1.93 (0.85–4.38) | |
| | Others | 26 (3.5) | 8 (1.1) | 1.91 (0.82–4.47) | 1.74 (0.69–4.41) | |
| Age at first sex | >15 years old | 359 (49.0) | 43 (5.9) | 1.00 | 1.00 | <0.001[b] |
| | ≤15 years old | 254 (34.7) | 76 (10.4) | 2.49 (1.66–3.73) | 2.12 (1.37–3.28) | |
| Parity | Nulliparous | 52 (7.1) | 10 (1.4) | 1.00 | 1.00 | 0.333 |
| | 1–2 births | 377 (51.3) | 57 (7.8) | 0.69 (0.33–1.45) | 0.83 (0.37–1.86) | |
| | ≥3 births | 187 (25.4) | 52 (7.1) | 0.54 (0.36–0.82) | 1.19 (0.51–2.82) | |
| History of STIs | No | 510 (69.4) | 84 (11.4) | 1.00 | 1.00 | <0.017[b] |
| | Yes | 106 (14.4) | 35 (4.8) | 2.01 (1.29–3.13) | 1.81 (1.11–2.95) | |
| Sexual partners | Single | 338 (46.0) | 36 (4.9) | 1.00 | 1.00 | <0.001[b] |
| | Multiple | 278 (37.8) | 83 (11.3) | 2.82 (1.85–4.30) | 2.42 (1.55–3.77) | |

[a]AOR, adjusted odds ratio; CI, confidence interval; COR, crude odds ratio; STI, sexually transmitted infection.
[b]Presence of statistically significant association ($P < 0.05$).

this is one of the few studies that have incorporated multiple facilities across Ethiopia, and it provides a much better picture for understanding the genotype diversity and its association with cervical abnormalities in Ethiopia.

The high prevalence of hr-HPV has serious consequences, with CC being the second most common cancer in women, especially in a country like Ethiopia with limited resources. Low screening coverage, combined with the high burden of oncogenic hr-HPV genotypes, which are not included in the current quadrivalent vaccine, puts many women at an increased risk of developing cervical precancer and cancer (24). Our findings support the reported and projected rise in CC incidence and mortality (25), highlighting the urgent need for better and feasible HPV testing access to enable early detection of precancerous lesions, in addition to national VIA screening, and address existing knowledge gaps regarding risk factors and preventive measures (26).

In the current study, the prevalence estimates are in line with those reported by other Ethiopian researchers in previous studies, reporting positivity rates of 13.1% (9) and 16.0% (27). These similarities might be due to the use of comparable study designs and the fact that the populations studied were health facility-based. In contrast, our results were lower than those of previous reports, where positivity reached 53% (11) and 45.3% (28). These differences may be due to differences in participant characteristics, detection methods, and study scope. These studies primarily focused on high-risk groups with gynecological complaints and suspected cervical abnormalities or HIV-positive women, which may have affected the prevalence estimates. In contrast, our study included multiple regions across Ethiopia, providing more representative data on hr-HPV distribution as a key factor influencing HPV prevalence.

The prevalence was higher than that seen in various Asian and European countries, in which HPV positivity is usually below 10%, such as Vietnam (3.5%) (29) and Germany (6.5%) (30). The lower positivity rate may result from the varying study

designs, populations, and measures taken to prevent and control CC. These studies were population-based and excluded high-risk participants from the analysis. A national plan in Vietnam was established to lower the CC burden, which may lower HPV infection risk factors, including HPV vaccination and women's education, with rigorous screening and treatment of cervical pre-cancer and CC. These efforts have probably helped lower infection rates and HPV positivity among the population. In contrast, our estimate was lower than that of many African countries, with a higher prevalence reported in the pooled estimate for Africa (45.3%) (28), sub-Saharan Africa (34%) (8), South Africa (28.5%) (31), and Burkina Faso (25.4%) (32). Differences in study areas, selection of study populations, sexual behaviors in the population, high prevalence of HIV and other STIs, and limited CC screening and prevention programs, such as lower vaccination coverage in the country, may explain the differences in these prevalence estimates. All these factors likely contribute to the higher prevalence.

In our study, HPV-16 was the predominant genotype. This is consistent with previous reports from different parts of Ethiopia, in which the most frequent genotype was HPV-16, with prevalence rates of 50.4% (33), 37.3% (34), 32.7% (35), and 31.8% (9). Our results are consistent with those of previous worldwide studies in Africa (13, 31). Most cases of CC and its associated cytological abnormalities are caused by HPV-16, and its genetic characteristics have allowed the virus to persist and induce high-grade cervical lesions (4). The frequent detection of HPV-16 in these studies suggests its principal contribution to CC. However, HPV-16 was not reported as the most common genotype identified in some countries; for example, HPV-68 was the most prevalent genotype in Kinshasa (36), HPV-16 was among the least common in Togo (37), and HPV-16 was not detected in Burkina Faso (32). This geographical variation in genotypes influences vaccine approaches, which, in turn, shape a country's broader prevention and control policies. Consequently, we need to establish a routine surveillance system to monitor genetic diversity and gain insight into its epidemiology, which will help guide and plan an effective prevention and control strategy for CC.

Our analysis indicates that women with multiple hrHPV infections exhibit distinct characteristics compared to those with single hrHPV infections. In this study, the prevalence of multiple HPV infections was higher in women aged 40–49 years (52.5%), compared with 45.5% in the same age group among women with single-type infections. This pattern suggests that multiple infections may persist in mid-adult women. Our findings align with evidence from a study conducted in China, which reported that the rate of multiple HPV infections was positively correlated with increasing age in the overall HPV-infected population (38). Moreover, when comparing key behavioral risk factors such as history of sexually transmitted infections (STIs) and lifetime number of sexual partners, the differences between women with multiple hrHPV infections and those with single-type infections were minimal. Regarding STI history, 27.5% of women with multiple infections reported a previous STI, compared with 30.4% among those with single-type infections. Similarly, 70% of women with multiple hrHPV infections reported having had multiple lifetime sexual partners, which was comparable to 69.6% among women with single-type infections. These findings suggest that, within this study population, STI history and lifetime number of sexual partners did not substantially differentiate women with multiple hrHPV infections from those with single-type infections.

The current study revealed that 65.6% of the women who tested positive for hr-HPV were infected with genotypes not targeted by the current Gardasil-4 (quadrivalent) vaccine, which has been in use in Ethiopia since 2018 and targets only HPV-6, HPV-11, HPV-16, and HPV-18. Importantly, we found that 30.9% of women with these non-vaccine genotypes had abnormal cytology results, indicating a clear risk of developing cervical abnormalities, thus strengthening the growing evidence of the increasing role of other hr-HPV genotypes in CC development. This finding underscores the urgent need for the Ethiopian Ministry of Health to evaluate current vaccination approaches and modify CC prevention protocols, especially to consider a broader type of vaccine,

incorporate genotypes not targeted in the quadrivalent, and encourage targeted CC screening initiatives based on identified genotype profiles to avoid the unnecessary burden of CC in Ethiopia. Previous Ethiopian studies have reported prevalence rates of 51.8% and 55% (35, 39). These findings indicate that several common HPV genotypes in Ethiopia are not covered by the current vaccination program, which is a critical gap in the program. If vaccine coverage targeting other hr-HPV genotypes is not expanded, HPV infections and CC will remain major public health issues in Ethiopia.

In the present study, HPV-35 has been reported as one of the dominant high-risk genotypes in the studied population, and this finding is in line with an African population study and has been strongly associated with cervical intraepithelial neoplasia and invasive cervical cancer (40). The absence of HPV-35 and other regionally common oncogenic genotypes in both the quadrivalent and the current nonavalent vaccine formulations underscores a critical gap in vaccine coverage for African populations. This finding reinforces the importance of continued regional human papillomavirus genotype surveillance to inform vaccination policies and guide future vaccine development tailored to genotype patterns circulating in Ethiopia and similar settings.

Furthermore, our findings underscore the clinical value of combining molecular hr-HPV testing with cytology for cervical cancer risk stratification. Women with hr-HPV positivity and atypical squamous cells of undetermined significance represent an intermediate-risk group with a higher likelihood of underlying precancer. In low-resource settings with limited colposcopy access, this combined approach enables earlier triaging, guides follow-up or treatment, and helps distinguish transient from persistent oncogenic abnormalities. Optimizing cervical cancer prevention in Ethiopia will therefore require both vaccination strategies targeting locally prevalent hr-HPV genotypes and screening algorithms that integrate molecular-cytology screening and continuation of VIA, which is routinely performed in most health facilities, to enhance early detection and risk-based management. Our study revealed a 5.7% prevalence of epithelial abnormalities in cervical sample smears. This lower rate may reflect the characteristics of the participants who visited health facilities for various reasons, but not necessarily for gynecological complaints. In contrast, higher prevalence rates have been reported in studies conducted in Ethiopia (13.1%) (14), Botswana (13.5%) (41), and South Africa (25.8%) (42). This is likely due to the study population included, where these women were recruited specifically from reproductive health and specialty clinics who visited the facilities for gynecological purposes and had high HIV positivity, as these factors are related to increasing HPV infection and persistence, which resulted in abnormal cytology results. In contrast, the prevalence of hr-HPV infection among these women was 40.5%. This is in agreement with findings of a study conducted in Ethiopia, which reported a rate of 38.7% (43). However, considerably higher prevalence rates have been documented, with 88% in another Ethiopian study (9). This wide variation is likely to stem from differences in the study populations, geographic settings, study periods, and severity of cytological results. For example, studies that included women with high-grade lesions or CC usually showed higher hr-HPV positivity, whereas community-based screening with mixed cytology grades tended to report lower estimates.

We examined the potential risk factors associated with hr-HPV infection and found that women reporting more than one lifetime sexual partner had more than two times the odds of infection compared to those with a single partner. Similarly, a history of STIs nearly doubled the risk of hr-HPV infection. This agrees with findings from other SSA studies (13), as well as studies conducted in Ethiopia (9, 26) and South Africa (31), which have shown that linking multiple partners and prior STIs significantly increases hr-HPV exposure and persistence. This highlights the need for better sexual health education, promotion of safe sexual behaviors and practices, and routine STI screening to reduce hr-HPV infection and related diseases. Similarly, women who experienced sexual debut at or before 15 years of age had a higher risk of hr-HPV infection than their counterparts. These findings align with previous Ethiopian and global studies highlighting the role of early sexual initiation in hr-HPV acquisition (13, 15). These findings underscore the

importance of interventions addressing behavioral and demographic risk factors and provide essential evidence to inform public health strategies for HPV prevention and CC control in Ethiopia. These findings highlight the importance of interventions targeting behavioral and demographic risk factors, which will lead us and provide crucial evidence to guide public health plans for better HPV prevention and CC control in Ethiopia.

## Strengths and limitations

Overall, the present study provides a comprehensive overview of the prevalence and genotype distribution of HPV and its association with cervical abnormalities among Ethiopian women, drawing on data from multiple health facilities. However, this study had some limitations. First, our study was conducted in health facilities, and there was a lack of longitudinal follow-up. We believe that it should be expanded to include community-level data to better assess persistent infection or lesion progression. Second, only government health facilities were included, and private clinics were excluded. Future studies should address these limitations through longitudinal follow-up and increase the sample size to include women from the general population.

## Conclusion

The overall positivity rate for hr-HPV infection in Ethiopia is high. Furthermore, a high proportion of hr-HPV that was not included in the Gardasil vaccine cocktail, the current vaccine used in Ethiopia, was identified. Women aged 40–49 years had an increased risk of hr-HPV infection. There was a significant association between early sexual activity before 15 years of age, multiple sexual partners, and a history of sexually transmitted infections. Hence, the introduction of a nonavalent vaccine and/or implementation of an early age screening strategy must be implemented for broader protection. A further periodic national surveillance strategy must be implemented to monitor genotype diversity.

## ACKNOWLEDGMENTS

We acknowledge Addis Ababa University and EPHI for their assistance. We also thank the participating health facilities and study participants. Finally, we thank Abbott Molecular Diagnostics and Addis Ababa University College of Health Sciences for providing essential PCR kits for hr-HPV testing.

## AUTHOR AFFILIATIONS

[1]Ethiopian Public Health Institute, Addis Ababa, Ethiopia
[2]Department of Microbiology, Immunology, and Parasitology, College of Health Sciences, Addis Ababa University, Addis Ababa, Ethiopia
[3]Ethiopian Federal Police General Hospital, Addis Ababa, Ethiopia
[4]Aklilu Lemma Institute of Health Research, Addis Ababa University, Addis Ababa, Ethiopia

## AUTHOR ORCIDs

Sisay Tadele  http://orcid.org/0000-0001-5526-3340

## AUTHOR CONTRIBUTIONS

Sisay Tadele, Conceptualization, Data curation, Formal analysis, Investigation, Methodology, Software, Validation, Visualization, Writing – original draft, Writing – review and editing | Amelework Yilema, Data curation, Formal analysis, Investigation, Software, Writing – review and editing | Belete Woldesemayat, Conceptualization, Data curation, Formal analysis, Investigation, Software, Supervision, Writing – review and editing | Kidist Zealiyas, Conceptualization, Data curation, Formal analysis, Investigation, Methodology,

Software, Supervision, Validation, Visualization, Writing – original draft, Writing – review and editing | Abinet Admas, Formal analysis, Investigation, Resources, Writing – review and editing | Tamrat Abebe, Formal analysis, Investigation, Project administration, Resources, Supervision, Writing – review and editing | Ayele Gebeyehu, Formal analysis, Methodology, Project administration, Resources, Supervision, Writing – review and editing | Gemechu Tadesse, Project administration, Resources, Supervision | Getachew Tollera, Conceptualization, Data curation, Formal analysis, Investigation, Methodology, Project administration, Resources, Supervision, Validation, Writing – review and editing | Nega Berhe, Conceptualization, Formal analysis, Investigation, Methodology, Project administration, Supervision, Validation, Writing – review and editing | Nigatu Kebede, Conceptualization, Formal analysis, Methodology, Supervision, Validation, Writing – review and editing

## DATA AVAILABILITY

The data are available from the corresponding author upon request.

## ETHICS APPROVAL

This study protocol was approved by the Research and Ethical Review Committee of the Aklilu Lemma Institute of Pathobiology, Addis Ababa University (reference number: ALIPB IRERC/157/2017/24) and the Ethiopian Public Health Institute (reference number: EPHI-IRB-558-2024). Institutional permission was obtained following approval and support letters from the IRB offices. Written informed consent was obtained from all the participants. Patient identifiers were de-identified. This study was conducted in accordance with the Declaration of Helsinki and its amendments.

## ADDITIONAL FILES

The following material is available online.

### Open Peer Review

**PEER REVIEW HISTORY (review-history.pdf).** An accounting of the reviewer comments and feedback.

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
