## [Reviewer comments · Microbiology Spectrum]

Microbiology Spectrum

High-risk human papillomavirus genotype distribution, cytological abnormalities, and associated factors among Ethiopian women: A multicenter study

Sisay Alamneh, Amelework Shiferaw, Belete Hailemariam, Kidist Shite, Abinet Alemneh, Tamrat Zeleke, Ayele Chernet, Gemechu Tadesse, Getachew Eticha, Nega Belay, and Nigatu Kebede

Corresponding Author(s): Sisay Alamneh, Addis Ababa University

Review Timeline:

Submission Date:	September 30, 2025
Editorial Decision:	November 10, 2025
Revision Received:	November 18, 2025
Accepted:	December 8, 2025

Editor: Rebecca Yee

Reviewer(s): Disclosure of reviewer identity is with reference to reviewer comments included in decision letter(s). The following individuals involved in review of your submission have agreed to reveal their identity: Paul Nagao (Reviewer #1)

Transaction Report:

DOI: <https://doi.org/10.1128/spectrum.03154-25>

Re: Spectrum03154-25 (*High-risk human papillomavirus genotype distribution, cytological abnormalities, and associated factors among Ethiopian women: A multicenter study*)

Dear Mr. Sisay Tadele Alamneh:

Thank you for the privilege of reviewing your work. Below you will find my comments, instructions from the Spectrum editorial office, and the reviewer comments.

Revision Guidelines

Sincerely,
Rebecca Yee
Editor
Microbiology Spectrum

Reviewer #1 (Comments for the Author):

Major comments:

1. Include more details in the Discussion. Please expand on the lack of coverage on prevalent HR-HPV specific to Ethiopia in the quad-valent vaccine. For example, there are studies that examine the impact of HPV-35 in sub-Saharan Africa (PMID: 38322502). It may also be helpful to clarify the combined impact of cytologic and HPV testing results on how the patients are

stratified for treatment. This allows readers to understand the implications of patients with a positive HR-HPV and ASC-US.

2. Define what does 'multiple infections' mean? Infected with something that's not HPV, other STDs?

3. Please include any longitudinal data that can help strengthen your results and discussion. For example, was there any data that clarifies whether participants were previously vaccinated for HPV, prior treatment, or any first time visits. Were the participants with multiple HR-HPV serotypes being screened for the first time?

4. Figure 2- fix 'category name' in the actual figure. Either write the exact name of the categories or remove it altogether and publish this as a colored figure given that your legend provided the name. Please include in the caption the abbreviation of the groups/categories. Ensure that the text in the figure (e.g. n%) is visible.

5. Expand on limitations of the study as needed. For example, if there is no longitudinal data for patients, then please include that as a limitation to the study.

Minor comments:

1. Put the location of the manufacturer right where it was first mentioned. For example, location of Abbott line 211-212 and Anyplex line 219 can be moved to lines 207-208.

2. It may add to the discussion to include the impact of vaccine cross-reactivity with certain HPV serotypes (PMID: 29074174).

3. Please add an explanation on why the inclusion criteria for age started at 30 years old vs 21 years old. Is it the standard of practice in Ethiopia? Is it due to the limitation of resources? Does your team believe that expanding the age range would add further insight to the results if a younger population is captured?

Reviewer #2 : Please see attachment for comments.

Paper Summary:

This study examines the prevalence of HR-HPV across eleven health facilities in Ethiopia and the lack of coverage certain HR-HPV in the quadrivalent HPV vaccine. The authors emphasize that their study provides more nuanced findings due to the inclusion of multi-regional facilities that provides a more general picture of HPV prevalence in Ethiopia. In addition, the study creates a call to action to examine the lack of certain HR-HPV serotypes specific to Ethiopia in the quadrivalent vaccine and HPV policy in Ethiopia. The authors do a commendable job of delving into this topic and this manuscript can be significantly improved with edits.

Major comments:

1. Include more details in the Discussion. Please expand on the lack of coverage on prevalent HR-HPV specific to Ethiopia in the quad-valent vaccine. For example, there are studies that examine the impact of HPV-35 in sub-Saharan Africa (PMID: 38322502). It may also be helpful to clarify the combined impact of cytologic and HPV testing results on how the patients are stratified for treatment. This allows readers to understand the implications of patients with a positive HR-HPV and ASC-US.
2. Define what does 'multiple infections' mean? Infected with something that's not HPV, other STDs?
3. Please include any longitudinal data that can help strengthen your results and discussion. For example, was there any data that clarifies whether participants were previously vaccinated for HPV, prior treatment, or any first time visits. Were the participants with multiple HR-HPV serotypes being screened for the first time?
4. Figure 2- fix 'category name' in the actual figure. Either write the exact name of the categories or remove it altogether and publish this as a colored figure given that your legend provided the name. Please include in the caption the abbreviation of the groups/categories. Ensure that the text in the figure (e.g. n%) is visible.
5. Expand on limitations of the study as needed. For example, if there is no longitudinal data for patients, then please include that as a limitation to the study.

Minor comments:

1. Put the location of the manufacturer right where it was first mentioned. For example, location of Abbott line 211-212 and Anyplex line 219 can be moved to lines 207-208.

2. It may add to the discussion to include the impact of vaccine cross-reactivity with certain HPV serotypes (PMID: 29074174) .

4. Please add an explanation on why the inclusion criteria for age started at 30 years old vs 21 years old. Is it the standard of practice in Ethiopia? Is it due to the limitation of resources? Does your team believe that expanding the age range would add further insight to the results if a younger population is captured?

Summary of the Manuscript

The manuscript presents an investigation into the prevalence and genotype distribution of hr-HPV among Ethiopian women, along with its association with cytological findings. The study goal is to contribute important data on hrHPV epidemiology in a region where such information is limited, which is of valuable significance to public health and cervical cancer prevention programs.

Overall, the topic is relevant and important. However, some issues require clarification and revision before the paper can be considered for publication. These include inconsistencies between figures, tables, and text; insufficient explanation of participant inclusion criteria; and unclear data presentation in key sections.

Major Comments

1. **Participant Inclusion Criteria:** It is not clear whether women with a previous positive cytology result or prior hrHPV infection were included in the study, and how this might have influenced the results. Please clarify how these participants were handled in the design and analysis.
2. **Inconsistency Between Figure 2, Table, and Text:** Figure 2 includes “ASC-H” cases; however, the corresponding data in the text and Table 2 does not discuss this category. This discrepancy needs to be resolved.
3. **HPV Types Among Women with Normal Cytology:** For participants with normal cytology, please specify which HPV genotypes were detected. How do these compare in prevalence or risk factors to those observed among women with abnormal cytology? This comparison would strengthen the epidemiological interpretation.
4. **Multiple hrHPV Infections:** The manuscript should further explore whether patients with multiple hrHPV infections differ in risk factors, age category, or previous screening history compared with those with single-type infections. This analysis would provide valuable insight into possible risk stratification

Minor Comments

1. **Table 2: High-Risk HPV Prevalence and Genotype Distribution:** The numbers in the “Cytology diagnosis” section do not add up correctly. Please re-check calculations and ensure that total counts and percentages correspond to the described categories.
2. **Figure 2:** The figure is visually confusing and difficult to read. The category labels are unclear. If you prefer not to include textual labels directly, consider simplifying by using consistent colors and providing a clear legend instead.

Overall Recommendation: The study addresses an important topic with potential public health impact. However, the manuscript requires some revision to correct inconsistencies in data presentation and to clarify methodological aspects related to participant selection and risk factor analysis. Once these issues are addressed, the paper could make a meaningful contribution to HPV epidemiology.

Response to Reviewers

* Please note that all page and line references are based on the cleaned PDF version of the manuscript.

Reviewer #1

Paper Summary:

This study examines the prevalence of HR-HPV across eleven health facilities in Ethiopia and the lack of coverage of certain HR-HPV in the quadrivalent HPV vaccine. The authors emphasize that their study provides more nuanced findings due to the inclusion of multi-regional facilities that provide a more general picture of HPV prevalence in Ethiopia. In addition, the study creates a call to action to examine the lack of certain HRHPV serotypes specific to Ethiopia in the quadrivalent vaccine and HPV policy in Ethiopia. The authors do a commendable job of delving into this topic, and this manuscript can be significantly improved with edits.

Major comments:

1. Include more details in the Discussion. Please expand on the lack of coverage on prevalent HR-HPV specific to Ethiopia in the quad-valent vaccine. For example, there are studies that examine the impact of HPV-35 in sub-Saharan Africa (PMID: 38322502). It may also be helpful to clarify the combined impact of cytologic and HPV testing results on how the patients are stratified for treatment. This allows readers to understand the implications of patients with a positive HR-HPV and ASC-US.

Response:

Thank you for this valuable and insightful comment. We have now expanded the *Discussion* section to address the points raised. We included a detailed explanation of the limited genotype coverage of the quadrivalent HPV vaccine, which protects against human papillomavirus types 6, 11, 16, and 18 but does **not** include other high-risk genotypes that are highly prevalent in sub-Saharan Africa, particularly **HPV-35**. We highlighted evidence showing the important role of HPV- 35 in cervical disease in African populations and emphasized that its absence in both quadrivalent and nonavalent vaccines represents a critical gap for effective prevention in Ethiopia.

Response to Reviewers

Additionally, we clarified the clinical implications of combined HPV testing and cytology results. We explained that women with hr-HPV positivity and atypical squamous cells of undetermined significance cytology represent a higher-risk group for underlying precancerous lesions compared to cytology alone. In resource-limited settings, this combined result supports risk-based triage for closer follow-up, improving early detection and patient stratification for care. These revisions help contextualize the screening outcomes and strengthen the public health relevance of our findings (page 14-15 , line 380 - 410)

2. Define what does 'multiple infections' mean? Infected with something that's not HPV, other STDs?

Response:

Thank you for the important observation and comment. In this study, multiple infections refer to participants who were infected with more than one Hr-HPV genotype simultaneously, including double, triple, quadruple, quintuple, and sextuple infections. In general, it indicates the presence of more than one high-risk human papillomavirus genotype in a single participant. And now has been defined in the operational definition section (page 4, line 142- 145).

3. Please include any longitudinal data that can help strengthen your results and discussion. For example, was there any data that clarifies whether participants were previously vaccinated for HPV, prior treatment, or any first-time visits? Were the participants with multiple HR-HPV serotypes being screened for the first time?

Response:

Thank you very much for your valuable observation and comment. We have now included two additional variables from our questionnaire in Table 1 (Sociodemographic and health-related characteristics of the study participants in Ethiopia, 2025), which describe the participants' human papillomavirus testing history and human papillomavirus vaccination status (page 8 , found in last rows)

Response to Reviewers

4. Figure 2- fix 'category name' in the actual figure. Either write the exact name of the categories or remove it altogether and publish this as a colored figure, given that your legend provided the name. Please include in the caption the abbreviation of the groups/categories. Ensure that the text in the figure (e.g. n%) is visible.

Response:

Thank you for the important comment. The correction has now been made as suggested.

5. Expand on limitations of the study as needed. For example, if there is no longitudinal data for patients, then please include that as a limitation to the study.

Response:

Thank you for the insightful comment and suggestion. In the Strengths and Limitations section, we have already addressed this by stating that the absence of longitudinal follow-up is one of the limitations of our study (page 18, line 429-431) .

Minor comments:

1. Put the location of the manufacturer right where it was first mentioned. For example, the location of Abbott line 211-212 and Anyplex line 219 can be moved to lines 207-208.

Response:

Thank you for the important observation. The manufacturer's location has now been added as suggested (page 5, line 173, and line 175).

2. It may add to the discussion to include the impact of vaccine cross-reactivity with certain HPV serotypes (PMID: 29074174).

Response:

Thank you for the important recommendation. However, in this manuscript, all study participants were unvaccinated. Therefore, we agreed that it would be more appropriate to refrain from discussing cross-reactivity and vaccine efficacy, and instead focus on informing the academic community about the extent and type of non-vaccine-targeted HPV genotypes detected in the studied population.

Response to Reviewers

3. Please add an explanation on why the inclusion criteria for age started at 30 years old vs 21 years old. Is it the standard of practice in Ethiopia? Is it due to the limitation of resources? Does your team believe that expanding the age range would add further insight to the results if a younger population is captured?

Response:

Thank you for the valuable comment and question. The age criteria for inclusion in this study, ranging from 30 to 65 years, were guided by international and national recommendations. According to the World Health Organization guideline, routine cervical cancer screening for the general population should begin at 30 years of age (World Health Organization, *WHO guideline for screening and treatment of cervical pre-cancer lesions for cervical cancer prevention*, 2nd edition). In alignment with this, Ethiopia adopted the same age threshold in the *Guideline for Cervical Cancer Prevention and Control in Ethiopia, 2015*, which recommends initiating cervical cancer screening at 30 years. Therefore, the study population was defined based on these nationally endorsed and globally standardized recommendations (page 4 , line 123) .
(<https://iris.who.int/server/api/core/bitstreams/329a5f3d-b423-48b3-beb1-26ee1240coa3/content>).
(<https://www.iccp-portal.org/sites/default/files/plans/Guideline%20Eth%20Final.pdf>).

Reviewer #2

Summary of the Manuscript

The manuscript presents an investigation into the prevalence and genotype distribution of hrHPV among Ethiopian women, along with its association with cytological findings. The study goal is to contribute important data on hrHPV epidemiology in a region where such information is limited, which is of valuable significance to public health and cervical cancer prevention programs.

Overall, the topic is relevant and important. However, some issues require clarification and revision before the paper can be considered for publication. These include inconsistencies between figures, tables, and text; insufficient explanation of participant inclusion criteria; and unclear data presentation in key sections.

Response to Reviewers

Major Comments:

1. Participant Inclusion Criteria: It is not clear whether women with a previous positive cytology result or prior hrHPV infection were included in the study, and how this might have influenced the results. Please clarify how these participants were handled in the design and analysis.

Response:

Thank you for your valuable and constructive comment. We have now revised the manuscript to present the inclusion and exclusion criteria clearly. In line with your suggestion, we have explicitly stated that women with a history of cervical abnormalities, previous hysterectomy, or prior high-risk human papillomavirus infection were excluded from the study (page 4, line 130 - 132).

2. Inconsistency Between Figure 2, Table, and Text: Figure 2 includes “ASC-H” cases; however, the corresponding data in the text and Table 2 do not discuss this category. This discrepancy needs to be resolved.

Response:

Thank you for your constructive comment and observation. We have now revised the manuscript and have carefully addressed and resolved the noted discrepancies. In addition, we included it in the text too (page 9, line 266 - 267).

3. HPV Types Among Women with Normal Cytology: For participants with normal cytology, please specify which HPV genotypes were detected. How do these compare in prevalence or risk factors to those observed among women with abnormal cytology? This comparison would strengthen the epidemiological interpretation.

Response:

Thank you for this important comment. We have revised the manuscript to specify the HPV genotypes detected in women with normal cytology findings as suggested, and the scope of the manuscript: “Within this group, HPV-16 accounted for 4.6% (32 cases),

Response to Reviewers

HPV-18, HPV-39, and HPV-51 each accounted for 0.7%, and dual HPV infections accounted for 2.9%.” All HPV genotypes (as single and multiple infections) detected in women with abnormal cytology were already presented in Table 3 with clear numbers and percentages (page 10, line 270 -272).

4. Multiple hrHPV Infections: The manuscript should further explore whether patients with multiple hrHPV infections differ in risk factors, age category, or previous screening history compared with those with single-type infections. This analysis would provide valuable insight into possible risk stratification

Response:

Thank you for this important comment and suggestion. We have now included a comparative analysis of women with multiple versus single hrHPV infections and key behavioral risk factors. “Our results show that multiple infections were more common in women aged 40–49 years (52.5% vs. 45.5%), consistent with evidence from a Chinese study showing a positive association between age and multiple HPV infections. Key behavioral risk factors showed minimal differences between groups: STI history (27.5% vs. 30.4%) and multiple lifetime sexual partners (70% vs. 69.6%). These additions have been incorporated as suggested (page 13, line 353 -366) .

Minor Comments

1. Table 2: High-Risk HPV Prevalence and Genotype Distribution: The numbers in the “Cytology diagnosis” section do not add up correctly. Please re-check calculations and ensure that total counts and percentages correspond to the described categories.

Response:

Thank you for the important observation. Your comments were accurate, and we have now corrected the errors and carefully checked all calculations to ensure accuracy (page 9 , in the last 4 rows of the table).

Response to Reviewers

2. Figure 2: The figure is visually confusing and difficult to read. The category labels are unclear. If you prefer not to include textual labels directly, consider simplifying by using consistent colors and providing a clear legend instead.

Response:

We sincerely appreciate your insightful comment, and the correction has been implemented as suggested (Figure 2).

Re: Spectrum03154-25R1 (***High-risk human papillomavirus genotype distribution, cytological abnormalities, and associated factors among Ethiopian women: A multicenter study***)

Dear Mr. Sisay Tadele Alamneh:

Your manuscript has been accepted, and I am forwarding it to the ASM production staff for publication. Your paper will first be checked to make sure all elements meet the technical requirements. ASM staff will contact you if anything needs to be revised before copyediting and production can begin. Otherwise, you will be notified when your proofs are ready to be viewed.

Sincerely,
Rebecca Yee
Editor
Microbiology Spectrum